# Hereditary Disorders of Manganese Metabolism: Pathophysiology of Childhood-Onset Dystonia-Parkinsonism in *SLC39A14* Mutation Carriers and Genetic Animal Models

**DOI:** 10.3390/ijms232112833

**Published:** 2022-10-24

**Authors:** Alexander N. Rodichkin, Tomás R. Guilarte

**Affiliations:** Brain, Behavior, and the Environment Program, Department of Environmental Health Science, Robert Stempel College of Public Health & Social Work, Florida International University, Miami, FL 33199, USA

**Keywords:** manganese, dystonia-parkinsonism, manganism, Slc39a14

## Abstract

Over the last decade, several clinical reports have outlined cases of childhood-onset manganese (Mn)-induced dystonia-parkinsonism, resulting from loss-of-function mutations in the Mn influx transporter gene *SLC39A14*. These clinical cases have provided a wealth of knowledge on Mn toxicity and homeostasis. However, our current understanding of the underlying neuropathophysiology is severely lacking. The recent availability of *Slc39a14* knockout (KO) murine and zebrafish animal models provide a powerful platform to investigate the neurological effects of elevated blood and brain Mn concentrations in vivo. As such, the objective of this review was to organize and summarize the current clinical literature and studies utilizing *Slc39a14*-KO animal models and assess the validity of the animal models based on the clinical presentation of the disease in human mutation carriers.

## 1. Manganese: A Double-Edge Sword—Essential Metal and Neurotoxicant

Manganese (Mn) is an essential trace element required for the proper function of many enzymes critical for human health [1,2]. However, chronic exposure to high levels of Mn has been known to be neurotoxic for nearly two centuries [3,4,5] and results in a neurological syndrome known as manganism [3,4]. Manganism is clinically expressed and defined as Mn-induced dystonia-parkinsonism [3,4,6]. Occupational and environmental exposures to high concentrations of Mn provided the principal cases of Mn-induced dystonia-parkinsonism in the early part of the 20th century [7,8]. However, contemporary occupational health practices and improvement in regulatory guidelines have significantly decreased human exposure to Mn in occupational settings resulting in a drastic decrease or absence of Mn-induced dystonia-parkinsonism cases resulting from occupational exposures [9]. From an environmental exposure perspective, certain regions of the world, including the United States, present with high levels of Mn in the environment, such as as naturally occurring high concentrations of Mn in the drinking water or Mn-contaminated air around metal refineries and mines. These environmental exposures have been shown to produce neurological deficits [3,4,7,8,10]. 

Furthermore, during the last two decades, clinical cases of Mn-induced dystonia-parkinsonism have been described in drug users from Eastern Europe and the Middle East. These users inject extremely high Mn concentrations in the form of potassium permanganate contaminated preparations of the home-made psychostimulant drug ephedrone [10,11,12,13]. Clinical conditions such as liver cirrhosis and the administration of high Mn-content in parenteral nutrition also highlighted cases of Mn-induced neurological disease [14,15]. 

More recently, a new genetic form of dystonia-parkinsonism associated with high blood and brain Mn concentrations has been described [6,16,17,18,19,20,21,22,23,24,25,26,27,28,29,30]. These cases are the result of inherited, autosomal recessive, loss-of-function mutations of the metal ion efflux and influx transporter genes *SLC30A10* and *SLC39A14*, respectively [6,16,17,18,19,20,21,22,23,24,25,26,27,28,29,30]. These new clinical cases have provided a wealth of new knowledge on the metabolism and neurotoxicology of Mn [6,16,17,18,19,20,21,22,23,24,25,26,27,28,29,30].

SLC39A14 is crucial to maintaining Mn homeostasis in the body. SLC39A14 is a Mn influx transporter protein that is highly expressed in the liver and the small intestine [6]. This specificity of tissue expression is hypothesized to be central to the function of the protein and its role in maintaining Mn homeostasis in the body. Mutations in *SLC39A14* result in Mn accumulation due to a lack of Mn absorption from the peripheral circulation and subsequent accumulation in the brain [6,22,23,24,25,26,27,28,29,30]. Although the cases of mutations in *SLC39A14* are rare, most mutations appear to affect the first ~150 amino acid residues of SLC39A14. Further breakdown of affected residues can be found in Figure 1. 

The recent availability of *Slc30a10* and *Slc39a14* global knockout (KO) mice and mutant *slc39a14* zebrafish have been useful preclinical models for studying the role of these transporters on Mn homeostasis and are now beginning to be used to discover the pathophysiology of hereditary disorders of Mn metabolism. In this review article, we will describe the current understanding and ongoing studies on the pathophysiology of Childhood-Onset Dystonia-Parkinsonism in *SLC39A14* mutation carriers and new knowledge gained from the availability of experimental animal models with global deletion of the *Slc39a14* gene. 

### 1.1. Childhood-Onset Dystonia-Parkinsonism in SLC39A14 Mutation Carriers 

Since 2016, an increasing number of clinical reports have described a progressive childhood-onset dystonia-parkinsonism resulting from inherited, autosomal recessive, loss-of-function mutations of the *SLC39A14* gene [6,22,23,24,25,26,27,28,29,30]. The timeline of most recent and significant discoveries can be found in Figure 2. *SLC39A14* encodes for a metal ion influx transporter of the same name. Originally, SLC39A14 was studied as a zinc (Zn) transporter; however, most recent studies and clinical cases indicate a major role in Mn transport and homeostasis [6,22,23,24,25,26,27,28,29,30]. In addition to Mn and Zn transport, SLC39A14 can also transport iron (Fe) and cadmium. In vitro studies have shown that when the *SLC39A14* mutations occur, although the protein is localized in the appropriate subcellular compartment—the plasma membrane—it is unable to effectively transport Mn. SLC39A14 is primarily expressed in the hepatocytes and is central to hepatic regulation of Mn homeostasis [6,32]. Therefore, individuals that express the mutated protein accumulate toxic levels of Mn in the blood and the brain due to the inability of hepatocytes to absorb Mn from the peripheral circulation. 

The marked increase in systemic Mn concentrations in mutation carriers increases the accumulation of Mn in the brain. The brain regions with high Mn accumulation include the basal ganglia, cerebellum (CB), brainstem (BS), and white matter with no change in blood levels of other essential metals such as Fe, Cu, and Zn [3,4,33]. Importantly, the regional brain accumulation of toxic Mn levels in the mutation carriers follows a pattern consistent with those observed in acquired occupational or environmental Mn exposures using T1-weighted Magnetic Resonance Imaging (MRI) [6,16,17,18,19,20,21,22,23,24,25,26,27,28,29,30,34]. No study to date has examined brain concentrations of Mn or other metals in postmortem brain tissue from *SLC39A14* mutation carriers. 

### 1.2. Clinical Presentation

First described by Tuschl and colleagues in 2016 [6], inherited loss-of-function mutations of the *SLC39A14* gene results in an early onset, progressive dystonia-parkinsonism. in which children miss developmental milestones and express progressive early onset dystonia, and bulbar dysfunction [6,22,23,24,25,26,27,28,29,30]. Within the first decade of life, and as early as 6 months of age, the mutation carriers exhibit severe generalized dystonia, spasticity, loss of ambulation, and, in some cases, parkinsonian features with hypomimia, tremor, and bradykinesia [6,22,23,24,25,26,27,28,29,30]. Currently, the etiological factor in the *SLC39A14* mutation-induced dystonia-parkinsonism is presumed to be the highly toxic concentrations of Mn measured in the blood and brain [6,22,23,24,25,26,27,28,29,30]. While these clinical populations have significantly advanced our understanding of Mn homeostasis and Mn-induced neurological disease, there is a paucity of knowledge on the neuronal systems and brain circuits that are affected. In addition, there is a lack of comprehensive studies on the pathophysiology of this new and debilitating pediatric genetic disease. 

### 1.3. Neuropathological Findings in Mutation Carriers

There is a paucity of knowledge on the neuropathology in *SLC39A14* mutation carriers and the limited information available comes from MRI brain imaging [6,22,23,24,25,26,27,28,29,30]. In the original publication by Tuschl and colleagues [6], it was noted that some of the *SLC39A14* mutation carriers express cortical and cerebellar atrophy, thinning of the corpus callosum, and widening of the extracerebral spaces. Another publication of cases by Martin-Sanchez and colleagues [25] supports the brain volume loss based on MRI imaging, including microcephaly, macrocephaly, and craniosynostosis. However, MRI imaging analyses of cases are limited in number and provide no detailed pathology of the neuronal systems and circuits that are affected. Finally, from a neuropathological perspective, there are no data that could be used to understand the neuronal cell types, circuits, and networks that are dysfunctional or degenerating, except for one case in which gross examination with H&E staining of the globus pallidus and dentate nucleus shows an apparent neuronal loss and gliosis [6] However, the neuronal phenotype(s) affected in this single case was not identified. Therefore, there is a complete lack of pathological studies in the brain of *SLC39A14* mutation carriers. 

### 1.4. Current Therapeutic Approaches in SLC39A14 Mutation Carriers

At present, the standard of care for *SLC39A14* mutation carriers remains limited in scope. Typically, one of the first pharmacological approaches in dystonia-parkinsonism cases consists of L-DOPA administration. L-DOPA is the precursor to dopamine (DA) that can pass through the blood–brain barrier and replenish brain DA concentrations in idiopathic Parkinson’s Disease (iPD) patients, where brain (caudate/putamen) DA depletion is the primary neurochemical hallmark of the disease [35,36]. However, *SLC39A14* mutation carriers are, in fact, refractory to L-DOPA therapies [6,22,23,24,25,26,27,28,29,30]. This is likely due to the lack of degeneration of dopaminergic neurons in the SNpc in the context of elevated brain Mn concentrations. 

Instead of L-DOPA, the current therapeutic approach consists of chelation therapy with CaNa_2_EDTA and dietary iron supplementation [6,22,23,24,25,26,27,28,29,30]. However, chelation therapy presents several caveats and challenges. First, CaNa_2_EDTA is not specific to Mn and will chelate other metals. Second, CaNa_2_EDTA therapy requires invasive infusions that will persist throughout the patients’ lives. Last, limiting dietary and water intake of Mn in these patients may prove to be exceedingly challenging, potentially offsetting the benefits of CaNa_2_EDTA therapy. Perhaps the biggest limiting factor stems from the current lack of understanding of the pathophysiology behind the disease and the course of the disease. For example, the most recently published case study describes a Korean patient whose *SLC39A14* genotype was determined in late adolescence [28]. Over the course of 5 years, CaNa_2_EDTA chelation therapy was successful at lowering serum and brain Mn concentrations, as confirmed by serum Mn concentrations and MRI brain imaging; however, the dystonia-parkinsonism symptomology did not significantly improve. This is important because the lack of neurological improvement associated with a decrease in brain Mn concentrations may suggest that the neurological deficits seen in mutation carriers and animal models are established early in life and require early intervention, as evidenced by the marked improvement shown in a 5-year-old patient from the original communication by Tuschl and colleagues [6,28]. In addition, some of the behavioral deficits may not be due to an increase in brain Mn levels alone, but rather due to a combination of cell-type-specific SLC39A14 dysfunction and Mn-induced toxicity [37]. Overall, our current understanding of the disease and its underlying pathophysiology is quite limited, and detailed investigations of the affected neuronal systems, circuits, and networks to develop mechanism-based therapeutic approaches are required.

## 2. Experimental Animal Model(s) Using Global *Slc39a14* Gene Deletion

As noted above, there is a lack of knowledge on the pathophysiology of the childhood-onset Dystonia-Parkinsonism in *SLC39A14* mutation carriers. A greater understanding of the neuronal systems and brain circuits that are affected could be useful in devising therapeutic strategies for this debilitating genetic pediatric disease and advance our understanding of acquired forms of Mn-induced dystonia-parkinsonism. Therefore, several laboratories have generated germline deletion of *Slc39a14* in murine models and characterized the behavioral phenotype. 

### 2.1. Behavioral Characterization

The first report of the *Slc39a19*-knockout (KO) mice behavioral phenotype was described in 2017 by Aydemir and colleagues [38]. *Slc39a14*-KO mice of both sexes were shown to have elevated blood and brain Mn concentrations with a small, but significant decrease in liver Mn concentrations consistent with those observed in *SLC39A14* mutation carriers [6]. Furthermore, there were no changes in serum Zn or Fe concentrations. In this study, motor function, gait, and balance were assessed at 8–16 weeks of age to characterize neurobehavioral changes in the *Slc39a14*-KO mice. Behavioral tests included open field spontaneous activity, balance beam traversal, hindlimb clasping, cylinder test, inverted grid, and pole descent test. The results indicate that 8–16-week-old *Slc39a14*-KO mice exhibited significant impairment in all measures of spontaneous activity including horizontal activity, total distance travelled, movement time, clockwise reversal, counterclockwise reversal, and margin time with a significant increase in resting time relative to wildtype (WT) mice. In the balance beam, *Slc39a14*-KO mice expressed a longer time to cross the beam, as well as a greater number of errors. Similarly, in the pole descent test, the *Slc39a14*-KO mice took a longer amount of time than WT. Finally, the hindlimb clasping score suggestive of dystonia-like movements was greater for the KO mice than WT and in the cylinder test. Interestingly, there were no differences between the KO and WT mice in the grip time using the inverted grid. Together, these findings indicate significant locomotor, coordination, and balance abnormalities consistent with what is observed in the human cases. 

Following the original report by Aydemir and colleagues [38], a different group developed and characterized global *Slc39a14*-KO mice [39]. In this study, the *Slc39a14*-KO mice also presented with highly elevated Mn concentrations in blood and in numerous brain regions, as well as in peripheral organs, except in the liver, resembling Mn accumulation in *SLC39A14* mutation carriers. Concentrations of Fe, Zn and Cu, in serum and several peripheral organs were measured. Except for serum Fe, which was significantly elevated in *Slc39a14*-KO mice, there were no changes in these metal concentrations in any of the analyzed tissues. From a behavioral perspective, the balance beam test, rotarod, and the CatWalk tests were performed to assess the behavioral phenotype. Unlike the study by Aydemir and colleagues [38], Xin and colleagues performed the assessment at different ages from 11 to 24 weeks. They found highly significant deficits in all motor function tests in the *Slc39a14*-KO mice relative to WT. However, it should be noted that, starting at 8 weeks of age, *Slc39a14*-KO mice presented with a significant decrease in body weight when compared to WT controls, which may have confounded the interpretation of the magnitude of the locomotor deficits. 

Another study by Jenkitkasemwong and colleagues [40] used global *Slc39a14*-KO male and female mice at 4 and 16 weeks of age to assess Mn homeostasis and motor function deficits. They found highly elevated Mn concentrations in whole blood as well as in multiple peripheral organs and brain regions except the liver and pancreas, where Mn concentrations were slightly decreased in *Slc39a14*-KO mice relative to WT. In this study, laser ablation ICP-MS was used to examine the regional brain Mn concentrations in the intact brains of *Slc39a14*-KO and WT mice. Jenkitkasemwong and colleagues showed that the highest concentrations of Mn in the brains of *Slc39a14*-KO mice was measured in the nucleus of the lateral lemniscus and parts of the cerebellum, with lower concentrations in parts of the globus pallidus, the dorsal raphe nucleus, and the pons, followed by the vestibular nucleus and the zona inserta. The findings obtained from the laser-ablation ICP-MS studies challenge the current dogma, which describe the highest brain Mn accumulation as being localized to the globus pallidus. Importantly, the loss of *Slc39a14* function did not affect Fe, Zn, or Cu concentrations in the brain. This study also showed that the regional brain distribution of Mn in WT mice was identical to the *Slc39a14*-KO mice but at a much lower level. This indicates that the loss of Slc39a14 affects (increases) the magnitude of the regional brain Mn concentration, but does not appear to affect the patterns of regional distribution. 

Motor function tests in this study included the pole descent test, beam-crossing and the rotarod [40]. Initially, in the motor function tests, WT and *Slc39a14*-KO mice were compared at 12 and 52 weeks of age. The results of the behavioral analysis indicate motor deficits of the KO mice in all behavioral tests. For example, in the pole descent test, the *Slc39a14*-KO mice had a longer descent time than WT; the KO mice also slipped and fell more often in the beam crossing, and they stayed in the rotarod for shorter times than WT littermates. The authors also provide a crucial observational assessment of the animals. They found that most of the *Slc39a14*-KO mice developed torticollis and were less active in the cage, with difficulty in rearing. The most significant observation difference in the *Slc39a14*-KO mice was the righting reflex, in that nearly all *Slc39a14*-KO mice failed the task. 

More recently, our laboratory has described metal concentrations and behavioral and neurochemical characterization of *Slc39a14*-KO male and female mice at postnatal day 60 (PN60) [41]. We found that both male and female mice had highly elevated whole blood and brain (striatum) Mn concentrations with no change in Fe, Cu, or Zn, consistent with what has been observed in *SLC39A14* mutation carriers and in other *Slc39a14*-KO mice studies from other groups. From a behavioral perspective, analysis of locomotor behavior in PN60 animals using automated activity cages during the dark cycle found highly significant impairments in distance traveled, speed, and vertical counts (rearing), with an increase in resting time in the *Slc39a14*-KO male and female mice relative to sex-matched WT controls. Analysis of motor coordination and balance using the rotarod found that PN60 *Slc39a14*-KO mice of both sexes fell from the rotating rod quicker than sex-matched WT, indicative of impairment in motor coordination and balance. Observational studies of dystonia-like movement using the tail suspension test were videotaped and analyzed by a blinded observed. The tail suspension test indicated that *Slc39a14*-KO mice expressed dystonia-like movements such as truncal and cervical torsion and hind and front limb clasping at a greater proportion of the time than WT animals. Importantly, and as previously noted [42], some of the *Slc39a14*-KO mice also expressed torticollis (lateral head tilt), retrocollis (backward head tilt), and straub tail. We also found that some of the *Slc39a14*-KO mice at weaning (PN21) express a backward walking phenotype, which persisted in the older PN60 *Slc39a14*-KO mice. The *Slc39a14*-KO mice expressed a wider separation of hindlimbs and a shuffling gait. Additionally, when the KO mice attempted to walk forward, there was a shuffling gait with dragging of the hind limb [41]. In this study, we measured no significant differences in body weight in either sex between WT and *Slc39a14*-KO mice, providing further evidence that the effects of the global deletion on behavior is a direct effect of changes in the neuronal system and brain circuits and networks and are not confounded by changes in body weight. 

Finally, a study by Giraldo and Janus examined the possibility that the postural abnormalities of torticollis and straub tail could confound the analysis of locomotor behavior in *Slc39a14*-KO mice [42]. However, they demonstrated that *Slc39a14*-KO mice that expressed torticollis and straub tail had test scores of locomotor behavior and coordination comparable to *Slc39a14*-KO mice that did not express torticollis and straub tail. Collectively, *Slc39a14*-KO mice express a behavioral phenotype that is consistent and relevant to the complex neurological phenotype described in early-onset dystonia-parkinsonism in *SLC39A14* mutation carriers. 

### 2.2. Pathophysiological Characterization of Slc39a14 Knockout Mice

As noted above, there is a paucity of knowledge on the neuronal systems, circuits, and networks that are affected in the *SLC39A14* mutation carriers. Similarly, none of the preclinical studies described above except our recent publication [41] have examined neurochemical or neuropathological changes in the brain of *Slc39a14*-KO mice. In our studies, we initially focused on the nigrostriatal DAergic system because it has been implicated in parkinsonism and dystonia [36]. In our published studies [41] to assess the nigrostriatal DAergic system in *Slc39a14*-KO mice, we aimed to interrogate various aspects of the dopaminergic circuitry. It is known that DA neuron cell bodies located in the substantia nigra pars compacta (SNpc) project their axons to the striatum, a basal ganglia structure that contains the highest concentration of DA in the brain. These DAergic neurons in the SNpc are the neurons that are susceptible to degeneration in iPD [36]. Therefore, we took different approaches to assess the structural integrity of DAergic neurons by analyzing DA neuron density, number, and soma size in the SNpc as well as DAergic terminal integrity in the striatum. We also measured the concentration of DA and its metabolites in the striatum. From a functional perspective, we assessed the basal and potassium-stimulated release of DA in the striatum using in vivo microdialysis. 

Our results in WT and *Slc39a14*-KO male and female mice at PN60 indicated that there were no differences in the concentration of DA or metabolites in the striatum of *Slc39a14*-KO mice of either sex relative to WT. Furthermore, there were no differences in the metabolite to DA ratio as a measure of DA turnover [41]. From a structural perspective, tyrosine hydroxylase (TH) immunohistochemistry in the striatum indicated no significant differences in DA neuron terminal staining between WT and *Slc39a14*-KO mice indicative of normal integrity of DAergic terminals in the striatum of *Slc39a14*-KO mice. This was followed by an analysis of TH-positive DA neurons in the SNpc using unbiased stereological cell-counting methods. For the latter, we quantified three different parameters: TH-positive neuron density, total neuron number, and soma size [41]. Consistent with the lack of DA terminal degeneration, we found no significant differences in TH-positive neuron density, total number of TH-positive neurons, or soma size in *Slc39a14*-KO mice of either sex relative to WT. To confirm our results using a different neuronal marker, we also performed the same type of studies using Nissl stain [41]. The analysis showed the same results as with TH staining: global deletion of *Slc39a14* and highly elevated Mn concentrations had no effect on the structural integrity of DAergic neurons in the SNpc. 

To assess the function of nigrostriatal DAergic neurons in the SNpc, we performed in vivo microdialysis in the striatum under basal and potassium-stimulated conditions. In this method, a cannula guide is implanted at the appropriate brain coordinates that will allow for the introduction of a semi-permeable dialysis probe to sample the extracellular fluid for analysis of DA levels. Furthermore, the cannula placement is confirmed using Nissl staining once the animals are euthanized. The concentration of DA in the extracellular fluid is analyzed using HPLC with electrochemical detection. Vesicular release of DA is stimulated using infusion of high-potassium chloride artificial CSF. Our studies showed that, in *Slc39a14*-KO mice, basal DA concentrations in the extracellular fluid were the same between WT and *Slc39a14*-KO mice. However, we measured a marked inhibition (85–90%) of DA release under potassium-stimulated conditions in *Slc39a14*-KO male and female PN60 mice relative to WT. 

To further assess the progression of the disease, we repeated our pathophysiological characterization of *Slc39a14*-KO and WT animals in an aged cohort at postnatal day 365 (PN365) [43]. To our knowledge, no other study has performed a detailed neurochemical and neuropathological characterization of aged *Slc39a14*-KO mice or animals that have been exposed to high concentration of Mn using a similar life-course approach. Consistent with the evidence from our earlier work [41], the aging *Slc39a14*-KO animals demonstrated a behavioral phenotype of decreased locomotor activity and dystonia. Our findings in PN365 *Slc39a14*-KO animals indicate an absence of neurodegeneration or neurochemical changes in the dopaminergic system. However, similarly to the PN60 cohort, we discovered marked inhibition of striatal DA release under potassium stimulated conditions in *Slc39a14*-KO animals when compared to age- and sex-matched WT controls [43]. Overall, these findings indicate that nigrostriatal DAergic neurons in *Slc39a14*-KO mice that exhibit highly elevated levels of brain Mn and significant locomotor impairment, and dystonia-like movements are structurally intact and are not degenerating. On the other hand, there is a highly significant impairment in their function in that there is marked inhibition of potassium-stimulated DA release. The latter is consistent with previous studies from our laboratory and others in which animals chronically exposed to high concentrations of Mn were found to have Mn-induced inhibition of DA release [41,43,44,45,46]. 

### 2.3. Loss of slc39a14 in a Zebrafish Animal Model

In the original publication of the early-onset dystonia parkinsonism human cases with *SLC39A14* inherited mutations, Tuschl and colleagues [6] described the generation of a zebrafish *slc39a14* null mutant using CRISPR/Cas9 genome editing [6]. Homozygous *slc39a14* zebrafish exhibited a 35% increase in Mn levels at 5 days post fertilization (dpf) [37] and 72% at 14 dpf. Importantly, like human mutation carriers and *Slc39a14*-KO mice, levels of Fe and Zn were not affected. The exposure of WT and homozygous *slc39a14* mutants to Mn resulted in a much higher degree of Mn accumulation and increased sensitivity to Mn-induced toxicity. Analysis of locomotor activity in the mutant zebrafish showed that they expressed a prominent decrease in their average locomotor activity [6,37]. The authors note that this effect of the highly elevated levels of Mn in the mutant zebrafish with reduced locomotor activity could be associated with impaired DA neurotransmission in zebrafish [37]. In summary, this work indicates that the zebrafish *slc39a14* mutants recapitulate some aspects of the *Slc39a14*-KO mice and the human mutation carriers. 

In a subsequent publication by the same investigators, they describe studies in zebrafish *slc39a14* mutants in which they performed transcriptomics analysis of individual mutant larvae and unaffected siblings to identify novel targets of Mn toxicity [37]. Analysis of differentially expressed genes from four different exposure conditions resulted in three groups of genes with distinct expression profiles. The first group was associated with differentially expressed genes in Mn-exposed siblings compared to their non-exposed siblings, a group that was called “Mn toxicity”, which is independent of genotype. The second group was associated with genes that showed increased sensitivity to Mn in slc39a14 mutant zebrafish defined, as differentially expressed in mutant zebrafish that were exposed to Mn compared to their mutant siblings that were not exposed. This group was named “increased sensitivity of mutants to Mn”. Finally, the third group was composed of genes that were differentially expressed in unexposed mutants compared with their unexposed siblings (normal), and this group was named “mutant effects”. 

An examination of the differentially expressed genes under the three categories, (1) Mn toxicity, (2) increased sensitivity of mutants to Mn, and (3) mutant effect, led to two groups of genes that were common to all three categories. These are calcium homeostasis and presynaptic neurotransmitter release. In the calcium homeostasis group by function, there were several genes that were up- or downregulated. Importantly, in the presynaptic neurotransmitter release group by function, all genes were reduced in expression Table 1 and [37]. Therefore, disruption of the calcium-sensitive synaptic vesicle neurotransmitter release apparatus in presynaptic terminals appears to be a target of Mn-induced neurotoxicity and the loss of Slc39a14 function. These observations are consistent with our previous findings that *Slc39a14*-KO mice [41,43], as well as Mn-exposed non-human primates [44,45], have a marked inhibition of DA release in the striatum. Future studies should determine if the effect on neurotransmitter release is specific to DA or other neurotransmitter systems are also affected. 

The study by Tuschl and colleagues, [37] using non-exposed *slc39a14* mutants and controls also provides important information about the impact of the mutation alone in the absence of Mn exposure. They show that several groups of genes by function are affected in the mutant zebrafish that do not express high levels of brain Mn. The authors suggest that the mutation of *slc39a14*, a Mn influx transporter, results in a functional Mn deficiency in yet to be identified brain cells and most of these gene changes can be recue by supplying Mn to the mutant zebrafish [37]. Therefore, it is proposed that the loss of function of *SLC39A14* in humans or *Slc39a14* (*slc39a14*) in mice (zebrafish results in simultaneous high levels of Mn and a Mn deficiency. This is an important concept that needs to be confirmed in mouse models. 

## 3. Conclusions

In summary, the recent advances in understanding the role of SLC39A14 function in modulating Mn homeostasis and neurotoxicity has provided novel information of Mn-induced dystonia-parkinsonism. In both humans and experimental animal models, our understanding of alterations in neurotransmitter systems and circuits is severely lacking. Further studies will allow for novel therapeutic strategies based on rigorously defined mechanisms of disease to be advanced. Furthermore, the neurological deficits of the human cases are by no means complete. In this context, novel findings in animal models can provide the basis for testing additional neurological and sensory function in human cases.

## Figures and Tables

**Figure 1 ijms-23-12833-f001:**
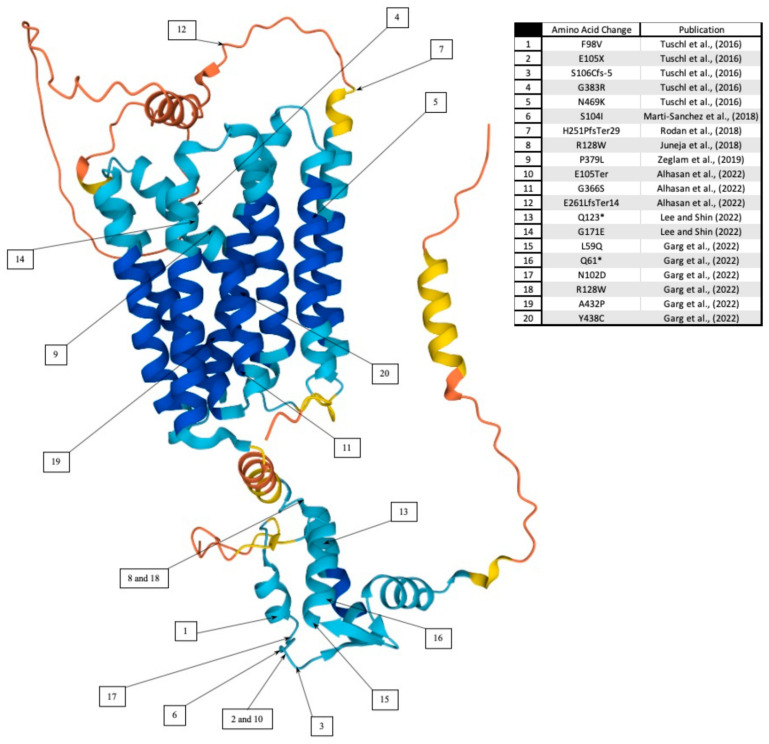
The protein structure of SLC39A14. The 3D structure was obtained using AlphaFold [31] (AF-Q15043-F1). Corresponding mutations and related work are indicated within the accompanying number table. Color gradient indicates model confidence (blue > 90; cyan 70–90; yellow 50–70; orange < 50). (See Refs. [6,24,25,26,27,28,29,30]).

**Figure 2 ijms-23-12833-f002:**
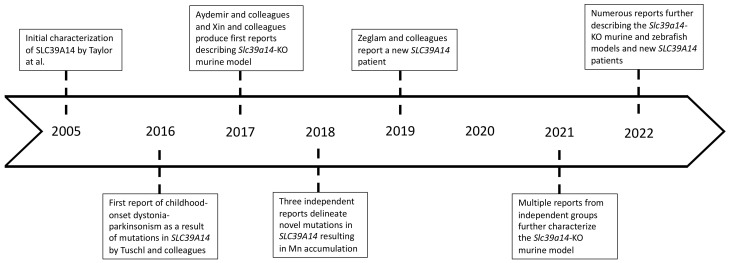
Timeline of significant discoveries related to *SLC39A14* human cases and disease models.

**Table 1 ijms-23-12833-t001:** Neurobehavioral assessments and their outcomes in Slc39a14-KO mice.

Publication	Neurobehavioral Assessment	Outcome
[38]	Spontaneous open field locomotor activity	KO animals present with a decrease in open field locomotor activity when compared to WT controls
	Balance beam traverse	KO animals required more time to cross the balance beam with more errors
	Pole descent test	KO animals exhibited an increase in the amount of time required to descend the pole
	Hind limb clasping reflex	Impairment of the hind limb clasping reflex function in KO animals
	Inverted grid assay	No difference detected between WT and KO animals
[39]	Balance beam traverse	KO animals present with balance beam traverse performance beginning at 11 weeks of age
	Rotarod	Significant decrease in maximum speed when compared to WT controls
	CatWalk Gait Analysis	Irregularities in gait patterns with longer swing speeds in right forelimb and left hindlimb
[40]	Pole descent test	KO animals showed an increase in the overall latency to descend. Low diet Mn treatment did not have a significant effect on PDT performance
	Balance beam traverse	KO mice slipped and/or fell at a higher frequency than the WT counterparts. Overall time to cross the beams was not significantly different between genotypes
	Rotarod	KO animals exhibited shorter latencies to fall off the rotating rod, relative to weight-, sex- and age-matched WT controls. Low-Mn diet did not rescue the phenotype
	SHIRPA Screen	No differences were detected between genotypes using the following parameters: tremors, respiration, transfer arousal, palpebral closure, piloerection, pinna reflex, limb grasping, wire maneuver, negative geotaxis. Diet did not have a significant effect on the outcomes.
	Qualitative Observations	KO animals appeared to be less active than the WT counterparts when compared to the WT controls in a large, open cage.
[42]	SHIRPA Screen	Presence of torticollis in KO animals did not affect their scoring on the SHIRPA Screen, when compared to KO counterparts with no torticollis
	Pole descent test	Torticollis in KO animals did not have a significant effect on PDT performance
	Balance beam traverse	Torticollis in KO animals did not have a significant effect on balance beam traverse performance
	Rotarod	Torticollis in KO animals did not have a significant effect on latency to fall on the rotarod
[41]	Open field locomotor activity	KO animals presented with a decrease in overall distance travelled, speed and rearing activity when compared to age- and sex-matched WT controls. Resting time was increased.
	Rotarod	Latency to fall was decreased in KO animals when compared to WT controls
	Tail suspension test	On average, KO animals exhibited more instances of hind and front limb clasping, truncal and cervical torsion. Together these behaviors are interpreted as dystonia-like movements.
	Qualitative Observations	At post-natal day 21, KO a proportion of KO animals exhibited backward walking, which persisted until post-natal day 60.
[43]	Open field locomotor activity	KO animals presented with a decrease in overall distance travelled, speed and rearing activity when compared to age and sex matched WT controls. Resting time was increased.
	Rotarod	Latency to fall was not significantly different from age and sex matched WT controls. However, this is likely due to a decrease in performance of the WT animals.
	Tail suspension test	On average, KO animals exhibited more instances of hind and front limb clasping, truncal and cervical torsion.

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
