# Peer review of "Hereditary Disorders of Manganese Metabolism: Pathophysiology of Childhood-Onset Dystonia-Parkinsonism in SLC39A14 Mutation Carriers and Genetic Animal Models"

_ijms, 2022, doi:10.3390/ijms232112833_

Round 1

Reviewer 1 Report

The review of Rodichkin & Guilarte, is a very interesting metanalysis on SLC39A14- Dystonia-Parkinsonism. This is a Hot Topic in the dystonia field and an old story concerning Parkinson's and Parkinsonism.

The paper is well structured, clear, linear, and comprehensive of the most recent literature evidence. 

The authors that are involved in the dystonia-parkinsonism field are more competent. They did not report their more recent work (PMID: 36152728) 

I would suggest including it for the sake of completeness.

The table completed in the additions finds its proper placement as additional material.

However, I would suggest including a summary graph (timeline) of the most relevant events they discussed; and a figure of the gene or the metal transporter with its most known mutations

Author Response

We would like to thank Reviewer 1 for the thoughtful comments and feedback.

We incorporated all the suggestions, which can be found in the revised manuscript.

Reviewer 2 Report

 The function of SLC39A14  in  maintaining Mn homeostasis in the body is adequately described by the authors; with   childhood-onset dystonia-parkinsonism resulting from inherited, autosomal recessive,loss-of-function mutations of the SLC39A14 gene . The current therapeutic approaches in SLC39A14 mutation carriers with CaNa2EDTA and dietary iron supplementation  instead of levodopa has been discussed. The various  experimental animal models   using global Slc39a14 gene deletion to study the neurobehavioral assessments and their outcomes in Slc39a14-KO  in understanding the role of SLC39A14 function in modulating Mn homeostasis and neurotoxicity has provided important information which can help up to unravel the pathophysiology of this challenging disorder have been discussed in details .The results of animal experiments could possibly  be extrapolated to humans .   induced dystonia-parkinsonism. 

Author Response

We appreciate the Reviewer 2 kind comments and feedback. In regards to the comment that animal experiments could possibly be extrapolated to humans with dystonia-parkinsonism, we have done that throughout the manuscript.